# The Use of Oncept Melanoma Vaccine in Veterinary Patients: A Review of the Literature

**DOI:** 10.3390/vetsci9110597

**Published:** 2022-10-28

**Authors:** MacKenzie A. Pellin

**Affiliations:** School of Veterinary Medicine, University of Wisconsin-Madison, 2015 Linden Drive, Madison, WI 53706, USA; mackenzie.pellin@wisc.edu

**Keywords:** melanoma, melanocytoma, DNA vaccine, comparative oncology, veterinary oncology

## Abstract

**Simple Summary:**

Oncept is a vaccine that is used to treat melanoma, a cancer of pigmented cells that can spread throughout the body, in dogs and other veterinary species. Its use is controversial, however, as the results of studies have been mixed as to whether it provides benefit and increases lifespan when used. This review summarizes the history of Oncept, and the various scientific studies that have reported on its use in dogs and other animals. While the vaccine appears to be safe, there does not appear to be evidence that it improves outcome when used. Further studies to better evaluate its use in patients, and exploration of other treatment options for melanoma, need to be performed.

**Abstract:**

The Oncept melanoma vaccine is xenogeneic DNA vaccine targeting tyrosinase. It is USDA approved for treatment of stage II to III canine oral melanoma and is also used off-label for melanomas arising in other locations and in other species. While the vaccine appears safe, the published data is mixed as to whether it provides a survival benefit, and the use of the vaccine is somewhat controversial in the veterinary oncology community. In this paper, the published literature describing the use of Oncept is reviewed and evaluated.

## 1. Background

Melanoma is a common neoplasm affecting companion animals, particularly dogs. Arising from melanocytes derived from the neural crest [1], it can manifest anywhere in the body. In the dog it is most commonly found as an oral tumor, tumor of the digit, or tumor of the skin. The biological behavior among these locations varies widely, from locally aggressive and highly metastatic oral melanoma to relatively benign dermal melanoma. Staging of disease is performed according to World Health Organization (WHO) parameters: stage I tumors are less than 2 cm, stage II tumors are 2–4 cm, stage III tumors are > 4 cm and/or have evidence of lymphatic metastasis, while stage IV disease has evidence of distant metastasis [2]. Besides stage, other prognostic parameters include presence of metastatic disease, lymphatic invasion, nuclear atypia, mitotic index, Ki67 index, and degree of pigmentation [3]. Melanoma in humans primarily presents as a dermal neoplasia and is considered malignant. Despite the variation in location, melanoma arising in animals is similar to its human counterpart in both molecular characteristics and biological behavior [4]. While local therapy such as surgery and/or radiation therapy is used as the primary therapy to target regional disease, systemic metastases for non-dermal locations represents a therapeutic challenge as multiple studies have demonstrated relative resistance to systemic chemotherapy with reported response rates of 8–28% and no evidence of improvement in survival [5,6,7].

While systemic chemotherapy has failed to improve outcomes, immunotherapy has shown promise in both humans and companion animals [4,8,9]. Among the first indications that melanoma represented an immune responsive disease were anecdotal reports of spontaneous remission in multiple species [10,11]. Investigated immune therapies to combat melanoma include those targeting the innate and adaptive immune responses. Techniques to non-specifically induce the innate immune response in dogs have included use of Corynebacterium parvum [12], liposome-encapsulated muramyl tripeptide phosphatidylethanolamine (L-MTP-PE) with or without granulocyte macrophage colony stimulating factor (GM CSF) [13], and PEGylated tumor necrosis factor (TNFα) [14], amongst others [4,9]. Techniques to target the adaptive immune system have included injection of autologous dendritic cells expanded ex vivo via transduction with human gp100, a melanoma antigen, as an adjuvant to radiation therapy [15], as well as allogeneic whole-cell tumor vaccines expressing various melanoma antigens [16]. Adoptive cell transfer, utilizing transfusion of tumor specific T lymphocytes to cancer patients, has shown success treating human melanoma but has not yet been largely evaluated in canine melanoma patients [17]. Additional immunotherapy techniques previously evaluated have included monoclonal antibodies [18], targeting cancer-specific antigens, and oncolytic virotherapy, with previous studies utilizing Adenovirus or Canine Distemper Virus (CDV) against canine melanoma both in vitro and in vivo [19].

## 2. Development of Oncept

A novel approach was taken with the advent of the Oncept melanoma vaccine. This vaccine uses xenogeneic DNA to elicit an immune response in dogs; specifically human DNA encoding tyrosinase. Tyrosinase is an enzyme that is highly conserved amongst mammalian species in the melanin pigment synthesis pathway, catalyzing dihydroxyphenylalanine as a critical and rate limiting step [20]. Human DNA encoding tyrosinase is placed into a bacterial plasmid and injected into the canine patient [21]. This technique was adopted from an investigational vaccine developed at Memorial Sloan-Kettering Cancer Center and that was evaluated in human patients with advanced (stage III or IV) melanoma [22]. This study evaluated 18 patients and determined that the vaccine was safe and effectively induced immunity based upon T cell responses. The resultant human tyrosinase protein is at least 85% homologous (with some reports up to 92% homologous) to canine tyrosinase, therefore both different enough to elicit an immune response, and similar enough to provide an appropriate target in the canine melanoma cells [23]. Initial studies also evaluated murine tyrosinase with or without human granulocyte colony stimulating factor (GM-CSF), as well as murine GP75, a tumor associated antigen located within the membrane of melanosomes; based upon these studies, the most robust immune response and correlating improvement in survival was associated with use of human tyrosinase [24]. The vaccine is delivered via Biojector 2000 or Vet Jet carbon dioxide powered jet needleless transdermal delivery devices that are FDA approved for intramuscular injections. The plasmid is then collected by dendritic cells to begin the immune response. Humoral response post vaccine has been documented and antibodies were documented to persist for three to nine months [25]. However, in this study only three of nine vaccinated patients demonstrated robust humoral response. It was also noted that degree of response did not necessarily correlate to clinical response. A few dogs from initial studies did develop localized or more systemic depigmentation or vitiligo, which also suggests a systemic immune response targeting tyrosinase [24].

Based upon initial studies [24,25] demonstrating safety and humoral response, the vaccine was conditionally licensed by the USDA Center for Veterinary Biologics in 2007. Prescribed dosing was one injection containing 0.4 mL (102 µg of DNA) of vaccine administered intradermally by the Vet Jet injector (as seen in Figure 1) every two weeks for four doses. Patients that survived beyond the initial vaccine series were eligible for “booster” vaccines every six months.

The first clinical study evaluating efficacy of Oncept in canine patients evaluated 58 patients with WHO stage II or III, histologically confirmed oral melanoma [23]. In this study, locoregional disease control was achieved with surgery and radiation therapy if necessary based on incomplete margins. Patients receiving Oncept were prospectively enrolled at multiple veterinary oncology centers throughout the United States. No systemic toxicity was seen after administration; local toxicity included mild to moderate pain at injection site, cutaneous wheal formation, or rarely hematoma. No adverse events required veterinary intervention. To evaluate survival, the 58 Oncept treated dogs were compared to a historical control population of 53 dogs who achieved locoregional disease control via surgery. These control patients were derived from a single academic institution, and had participated in previous clinical trials either receiving a placebo or another treatment that was determined to not have significant anti-tumor activity after surgery. Control patients were matched to prospectively enrolled patients in terms of signalment and staging criteria as closely as possible. When comparing these groups, median survival time based upon Kaplan–Meier analysis was not reached for the Oncept treated group and 324 days for the historical controls. When the 25th percentiles of MSTs for the treatment group was calculated it was 464 days compared to 156 days for historical controls.

One of the major criticisms of this study is that the Oncept treated population was highly censored due to loss of patient follow-up or death due to other causes. In fact, only 15 dogs died due to melanoma disease and were therefore included in the Kaplan–Meier analysis. Reasons for censoring included loss to follow up (10 dogs), removal from the study due to pursuance of other treatment for melanoma or another cancer (9 dogs), death due to unrelated causes (16 dogs), or being alive at the end of the study period (8 dogs). It was not clear how many dogs that participated in the study did have necropsy performed at the time of death. It should be noted that three dogs did have evidence of recurrent or metastatic melanoma on necropsy; however, these were considered incidental findings and it was determined that these patients died to unrelated causes and were censored from survival analysis.

Other published criticisms [26,27] of this study have included the use of the historical, non-contemporaneous control group which may not accurately account for advances in local therapy, or in pathologic advances in the interpretation and assessment of melanoma, such as more routine use of mitotic index for assessment of biological behavior. Use of overall survival time for assessing outcome can be biased by differences in end-of-life decisions amongst clients, while the use of progression free survival (PFS) or disease-free interval (DFI) is a more objective measure of response, but difficult to evaluate in a retrospective setting. The non-randomized nature of the treatment intervention and the overall small number of patients are also limitations. Regardless, the results of this study helped Oncept become fully licensed by the USDA in 2010, qualifying as a veterinary biological product. USDA authorization only requires demonstration of safety and a reasonable expectation of efficacy [28]. In contrast, to qualify for FDA licensure as a cancer therapeutic, Oncept would have needed to have gone through more robust demonstration of efficacy including randomized and placebo-controlled clinical trials [29].

## 3. Retrospective Studies

After full-licensure and increasing clinical experience, and considering the criticisms outlined above, a number of retrospective studies emerged seeking to further report on the use of Oncept in canine patients with oral melanoma.

A study by Ottnod et al. retrospectively evaluated 45 dogs with locoregional control of oral melanoma achieved by surgery, combined with radiation therapy if necessary for incomplete margins [27]. Dogs with stage IV disease, macroscopic disease, or dogs receiving other systemic therapies were excluded. Twenty-two of the 45 dogs received Oncept as an adjuvant after locoregional treatment. Disease and patient characteristics were similar between patients that did and did not receive the vaccine. The PFS, DFI, and MST were not statistically significant between vaccinated (199 days, 171 days, 485 days respectively) and unvaccinated dogs (247 days, 258 days, 585 days respectively). The study also separately evaluated dogs with stage II or III disease, to better compare to the original Grosenbaugh study. The results of this sub-analysis still did not show a statistically significant difference in PFS or MST between patients receiving Oncept vs. those that did not receive further therapy (PFS: 179.5 days vs. 247.5 days, MST: 477 days vs. 491 days respectively). However, the DFI was significantly longer for patients not receiving the vaccine (331 days vs. 140 days (*p* = 0.02)). In this study, histopathology characteristics suggestive of less aggressive biological activity, such as mitotic index less than 4, nuclear atypia less than 30, and Ki67 less than 19.5, were suggestive of improved patient outcomes.

Another study by Treggiari et al., reported on the use of Oncept in the United Kingdom [30]. Thirty-two dogs with stage I to stage III oral melanoma were treated with surgery and received Oncept post-operatively. Seven patients also received radiation therapy (four fractions of 8–9 Gy weekly) if surgical excision was incomplete. Median survival time was 335 days with a median progression free survival time of 160 days; 20 dogs in this study, representing 62% of the patient population, died due to melanoma. A limitation of this study is that it did not include a control population, although survival times were similar to those reported in other studies. Another study out of the United Kingdom by Verganti et al., retrospectively evaluated 69 dogs across five institutions [31]. Dogs with well-differentiated oral melanomas, as determined by histopathology, or with melanomas of the haired lip were excluded. Patients who were not fully staged were also excluded, although full staging was not defined. Fifty-six patients were treated after locoregional control was achieved via surgery and/or radiation therapy, while 13 were treated in the macroscopic disease setting. For dogs with stage I to III disease, the MST was 455 days with a median DFI of 222 days. For the 13 dogs treated in the macroscopic disease setting, four complete responses (CR) and one partial response (PR) were observed, with three dogs having stable disease (SD); the remaining patients, including all dogs with stage IV disease, had progressive disease (PD). The MST for the patients with macroscopic disease was 179 days. Twelve dogs in this study did also receive systemic chemotherapy (carboplatin, mitoxantrone), metronomic chemotherapy (chlorambucil, cyclophosphamide), or toceranib.

Two multicenter studies have evaluated the use of Oncept as part of multi-modal therapy against oral melanoma. A study by Turek et al., evaluated 131 total dogs who received Oncept combined with surgery, radiation therapy, or both; no dogs in this study received systemic therapy besides Oncept, and there was no control population of dogs not receiving Oncept [32]. In this study eighteen patients (13.7%) did not complete the initial series of Oncept due to disease progression (*n* = 9) or due to loss to follow-up or non-compliance issues (*n* = 9); however, 50 dogs (38.2%) received at least one booster. For analysis, the patients were divided into three groups based upon presence of macroscopic disease (*n* = 54 dogs), microscopic disease (defined as incomplete surgical margins or residual metastatic lymph nodes after surgery, *n* = 15 dogs), and adequate local control (ALC) for dogs that achieved complete margins with surgery or had surgery followed by radiation therapy (*n* = 62 dogs). Median tumor specific overall survival for all dogs was 510 days, with median TTP of 304 days and PFS (including death as an event) of 260 days. Presence of macroscopic disease emerged as a significant negative prognostic factor in multivariable analysis, along with tumor size over 2 cm, lymph node metastases, and increasing stage (stage II or III). Adequate local control had a protective effect for overall survival (HR = 0.57) and addition of radiation therapy showed a protective effect for progression free survival (HR = 0.35). Another multicenter study by Boston et al. evaluating 151 dogs with melanoma of the oral cavity treated with surgical excision found no difference in survival for those patients receiving follow up systemic therapy (MST 335 days) compared to surgery alone (MST 352 days) [33]. However, only 14 dogs received the Oncept melanoma vaccine in this study, with a MST of 352 days. Other systemic therapies included maximum-tolerated dose chemotherapy (carboplatin, dacarbazine, lomustine, doxorubicin), metronomic chemotherapy, and another investigational melanoma vaccine. Twelve patients also received radiation therapy.

Finally, McLean and Lobetti documented the South African experience with Oncept after surgical excision in dogs with melanomas arising in various locations, including oral (*n* = 25), digital (*n* = 6), or infiltrative cutaneous (*n* = 7) melanomas [34]. At the end of the study period 6 dogs with oral melanoma were still alive at a median of 26 months, while the 16 dogs that had died of progressive disease had a median survival of 11.5 months (with 3 dogs dying due to unrelated disease). However, results are severely limited by small case numbers, lack of case details, and heterogeneous disease locations. Table 1 summarizes the currently published studies evaluating use of Oncept for treatment of oral melanoma in canine patients.

## 4. Use in Other Disease Sites

### 4.1. Digital Melanoma

While licensed for oral melanoma, given the lack of other treatment options for malignant melanoma there is significant interest in using Oncept as an off-label therapy. Various studies have evaluated use of Oncept in melanoma outside of the oral cavity. A 2011 study by Manley et al. evaluated 58 dogs who received the vaccine after diagnosis of melanoma arising from the nailbed and haired skin of the digit [35]. Fifty-seven dogs had digit amputation for local control; two dogs also received hypofractionated radiation therapy and three received carboplatin chemotherapy. The overall median survival time was 351 days from time of vaccination. Presence of lymph node or distant metastases was significantly associated with shorter survival with a MST 105 days compared to 533 days without metastasis. In the above study by McLean and Lobetti, the median survival for the six patients with digital melanoma receiving the vaccine was 36 months for the five dogs that survived to the end of the study period, and 12 months for the dog that died of disease progression [34]. Neither study included a control population of dogs who did not receive the vaccine, but previous literature suggests median survivals of 350–1350 days, and 1- and 2- year survival rates of 42–57% and 11–13%, respectively, for dogs with digital melanoma treated with surgery [36,37,38,39].

### 4.2. Dermal Melanoma

While melanomas arising from the haired skin are largely thought of as a relatively benign tumor in the dog, more aggressively behaving forms of cutaneous melanoma can exist. Oncept has been used as adjuvant therapy in these cases again due to lack of other systemic options. A 2018 multi-institution retrospective study evaluated 87 dogs with melanomas arising from the haired skin that were treated with surgical excision [40]. The median PFS and overall survival time were 1282 and 1363 days respectively. Patient age greater than 9.4 years and mitotic index greater than 20 were associated with decreased survival. Thirty-seven patients in this population received the Oncept vaccine with a median OST of 1282 days, which was not statistically different from patients receiving no adjuvant therapy after surgery. McLean and Lobetti also reported on seven dogs with melanoma arising from the haired skin; median survival was 22 months for patients surviving to the end of the study period with two dogs dying of disease at 2 and 6 months [34].

### 4.3. Other Sites

As melanoma can arise anywhere that pigmented cells reside [1], occasionally the disease presents outside of the “typical” locations. A recently published paper evaluated melanomas arising from the foot pad in 20 dogs [41]. Seven of these dogs received Oncept as adjuvant therapy after surgery, four as sole adjuvant therapy and three combined with chemotherapy (one patient each receiving gemcitabine, toceranib, or trametinib). Median survival time for all dogs was 240 days and 159 days for dogs receiving any adjuvant therapy. Melanoma arising from the anal gland seems similarly biologically aggressive with eleven dogs having a median PFS of 92.5 days and median survival of 107 days with various treatment modalities [42]. Three dogs received the Oncept vaccine after surgery, with one dog also receiving hypofractionated radiation therapy to metastatic sublumbar lymph nodes; survival for these three dogs was 155, 105, and 196 days.

## 5. Use in Other Species

Given the xenogeneic DNA mechanism to induce immune response utilized in Oncept, and with the highly conserved nature of tyrosinase [43] theoretically the vaccine should be effective across multiple species. Melanoma in cats is a rare disease but appears to be similarly biologically aggressive as in humans and dogs. Twenty-four cats received 114 doses of Oncept in a 2015 study evaluating safety [44]. Thirteen grade 1 or 2 adverse events were reported, which included pain at injection site, hyperpigmentation, muscle fasciculations, and mildly decreased appetite for 1–2 days after administration. The study unfortunately did not address efficacy of the vaccine or survival as the patients had melanomas arising from various disease sites and multiple and heterogeneous treatment modalities were used.

Cutaneous melanoma is common in horses, reaching prevalence of up to 80% in gray, adult horses. The majority are considered benign, however if untreated the majority will become more malignant in behavior [45]. The melanoma vaccine has also been used in an off-label basis in horses and has again been demonstrated to be safe [46]. A clinical trial evaluating use of Oncept in horses with melanoma is being performed, and while the results of this trial have not yet been published the initial reported results were promising with many patients having reported decreases in tumor size [45] The same group of researchers conducting the clinical trial have also evaluated immune response in healthy horses receiving the vaccine and have demonstrated increases in measurements of both humoral and cellular immunity post-vaccination [47].

Case reports for use of Oncept in zoo species also exist. Examples include use in a wild mountain Gorilla *(Gorilla beringei beringei)* [48] with melanoma arising from the left lip commissure and metastases to multiple lymph nodes, an African lion (*Panthera leo*) [49] with melanoma of the lip, and an African penguin (*Spheniscus demersus*) [50] with melanoma arising inside the nares.

## 6. Conclusions and Future Studies

Summarizing the currently available published data, there unfortunately fails to be definitive proof that the Oncept melanoma vaccine improves disease free intervals or survival in veterinary patients. It is unfortunate that the majority of currently available studies are retrospective in nature, given the inherent limitations that retrospective studies present including heterogeneous patient staging and treatment protocols, lack of randomization and control groups, and limitations to documenting patient outcomes [26,51]. While the study by Grosenbaugh was prospective in enrolling patients to receive Oncept, it was not randomized and used historical controls that were retrospectively collected for comparison and analysis. Given the limitations of the existing studies and current lack of overwhelming evidence of efficacy, a large-scale, prospective, randomized clinical trial should be performed. Of course, this is not an easy task and there are many hurdles to performing this type of trial in veterinary medicine, such as cost, patient recruitment, and ability to provide homogeneous staging and treatment [52].

Despite these limitations, there are documented cases of dogs with diffuse metastatic disease or macroscopic tumors disappearing after treatment with Oncept [21,24,30,32,46]. The reported cases of vitiligo also suggest systemic immune response to the vaccine [24]. Therefore, it is possible that on an individual basis Oncept may prove beneficial to some patients. However, these anecdotal cases should be viewed along with other documented cases of spontaneous remission, without treatment, in patients with melanoma [10,11].

Besides the largest question of efficacy, other smaller questions about Oncept’s use and monitoring remain. A persistent question is if Oncept is expected to be effective against amelanotic melanomas, in which melanin pigment is not being produced by the cancer cells due to de-differentiation [53]. A study by Smedley et al. demonstrated that only three out of 49 amelanotic melanoma histopathology samples were positive for tyrosinase with immunohistochemistry, and then only in 5–10% of cells [54]. In this same study, amongst the pigmented melanoma controls only six of the ten cases were positive for tyrosinase. However, other studies have suggested that tyrosinase is expressed in a majority of melanocytic tumors across species and have suggested that expression is not correlated with degree of pigmentation or biological behavior [43,55,56]. As we learn more about histopathological characteristics that predict biologic behavior in melanomas, such as mitotic index, this information will likely impact clinical decisions about treatment selection and need for adjuvant therapy. As an additional concern, it has been well documented that monitoring for immune therapies, such as Oncept, should not follow the same criteria as those used for more traditional cytotoxic therapy [57]. Treatment responses may be delayed in comparison to more traditional therapy, and a phenomenon called pseudo-progression can occur when the immune system, incited by the immunotherapy, reacts to the malignant cells, which leads to an inflammatory response. This can lead to the appearance of increased tumor size on imaging studies, which will ultimately be followed by lesion regression afterwards. Relying on RECIST criteria [58] in these situations may not be appropriate and could lead to abandonment of therapy when it is actually effective. However, this also can also lead to difficulty in assessing efficacy of treatment with immune therapies.

One conclusion that can undeniably be made is that Oncept is safe. Across all studies and in multiple species only low-grade local toxicity in a small number of patients has been reported [21,22,23,24,27,28,29,30,31,32,33,34,40,41,42,43,44,46]. There also regrettably remains a lack of other effective systemic therapy to address distant disease in this highly metastatic tumor which may drive oncologists to use Oncept, despite its questionable efficacy [5,6,7,8]. Given these facts, if a client owner is aware of the questions about Oncept’s efficacy and the limitations of the existing studies, and has the financial resources to dedicate to further treatment, it may still be an appropriate therapy to consider. However, further clinical trials evaluating Oncept, such as prospective, randomized clinical trials are still imperative to perform to advance scientific knowledge. Additional studies should be performed to better evaluate patient response to Oncept, such as evaluation of humoral or cell-mediated immune response, in an attempt to better understand and identify those patients who may have a positive response to Oncept. Beyond this, there continues to be a dire need for additional research evaluating therapeutic strategies to combat canine melanoma, such as further investigation of other immunotherapy directed strategies including checkpoint inhibitors which are now widely used in human patients with advanced melanoma [57].

## Figures and Tables

**Figure 1 vetsci-09-00597-f001:**
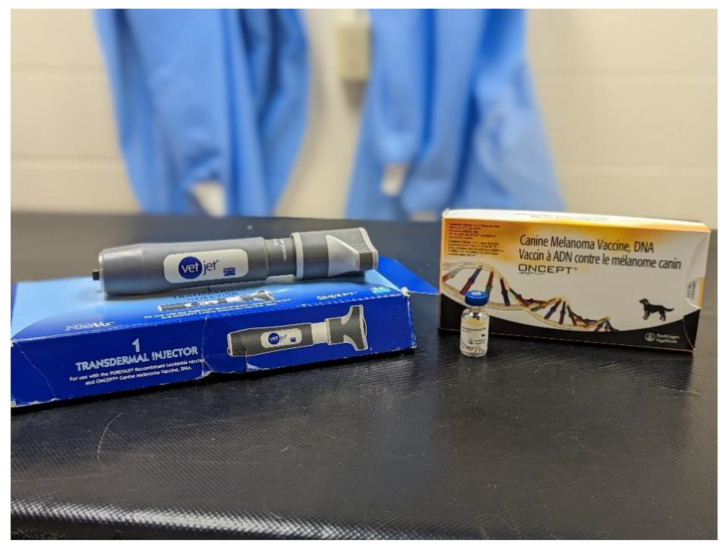
Photograph of Oncept melanoma vaccine drug box and vial, along with Vet Jet transdermal injector.

**Table 1 vetsci-09-00597-t001:** Summary of published studies evaluating the use of Oncept in canine patients with oral melanoma. Abbreviations: RT = radiation therapy, MST = median survival time, PFS = progression free survival, DFI = disease free interval, OS = overall survival, ALC = adequate local control.

Author	Type	Year	Dogs Receiving Oncept (N)	Control Dogs (N)	Other Treatment	Outcome	Comments
Grosenbaugh	Oncept cases prospectively enrolled; compared to historical control group receiving placebo or treatment with no significant antineoplastic activity	2011	58	53	Surgery +/− RT	MST not reached in treatment group vs. 324 days in controls 25th percentile MST: 464 days (Oncept) vs. 156 days (controls)	Heavily censored treatment population Non-randomized Historical controls
Ottnod	Retrospective	2013	22	23	Surgery +/− RT	Vaccinated vs. unvaccinated PFS: 199 vs. 247 days DFI: 171 vs. 258 days MST: 485 vs. 585 days	Subanalysis of stage II or III disease showed similar results
Boston	Retrospective	2014	14	137	151 dogs total received surgery; 12 received RT 32 received chemo 24 received immunotherapy (including Oncept)	Overall MST 346 days Oncept MST 352 days	No difference in survival between surgery alone vs. any systemic therapy
McLean & Lobetti	Retrospective	2015	25	n/a	Surgical excision	Dogs alive at end of study (*n* = 6), median of 26 months Dogs dead d/t PD (*n* = 16), MST 11.5 months	Study also evaluated digital and dermal melanomas
Treggiari	Retrospective	2016	32	n/a	All dogs had surgery, 7 also had RT	MST 335 days PFS 160 days	Stage I–III disease
Verganti	Retrospective	2017	69	n/a	56 dogs - surgery +/− RT 13 dogs macroscopic disease 12 dogs also received chemo	Locoregional control: MST 455 days, mDFI 222 days Macroscopic disease: MST 179 days; CR (4), PR (1), SD (3), PD (5)	Excluded patient with well-differentiated melanomas
Turek	Retrospective	2020	131	n/a	62 dogs ALC (surgery +/− RT) 15 dogs microscopic disease 54 macroscopic disease	TTP: 304 days PFS: 260 days Tumor specific OS: 510 days	Macroscopic disease negative prognostic factor; ALC, RT protective

## Data Availability

Not applicable.

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
