# Peer review of "The Use of Oncept Melanoma Vaccine in Veterinary Patients: A Review of the Literature"

_vetsci, 2022, doi:10.3390/vetsci9110597_

Round 1

Reviewer 1 Report

Well written! Minor edits:

1. Line 30: add "for non-dermal locations" after "systemic metastases"

2. Line 33: "immunotherapy" instead of "immune therapy"

3. Line 36: I think reference 10 is appropriate here, but reference 11 cites a case report from 2018... hardly a first indication of immune responsiveness (i.e. chronologically not an appropriate reference). Here's a better reference: Kalialis, L. V., Drzewiecki, K. T., & Klyver, H. (2009). Spontaneous regression of metastases from melanoma: review of the literature. Melanoma research, 19(5), 275–282. https://doi.org/10.1097/CMR.0b013e32832eabd5

4. Line 68: "correlate"

5. Line 117: Do we need "it should be noted that"? Leading the witness...

6. Line 125: A retrospective study can't "further evaluate" the use of Oncept. That would be the job of a prospective study, ideally a RCT. I understand the criticisms of the original data set, but a group of shoddy retrospectives are no better... I think this section should start with paragraph 2. 

7. Line 162: remove "larger", these aren't actually large

8. Line 216: "melanomas arising from the haired skin "are...""

9. Line 275-277: Completely agree - maybe the author can spearhead this effort?

10. Line 285: Please edit these references to reflect the earlier comment.

11. Line 316: While I completely agree with you, I don't know that this belongs in a review. I appreciate the statement of facts earlier in the manuscript, and appreciate (and can read between the lines) the text of lines 313-316.

Reviewer 2 Report

The paper provided by MacKenzie A. Pellin refers to the use, efficiency, and safety of the Oncept melanoma vaccine. The submitted article is interesting and can be useful for veterinary clinical oncologists. The author has made a fair and detailed review of the available studies on the Oncept vaccine. The author draws attention to the need for a prospective study and the immunohistochemical analysis of tyrosinase expression in patients' tumor tissues before the therapy.

In the opinion of the reviewer, the article may be improved with some changes.

1.      The described vaccine is a human tyrosinase encoding DNA-based approach. Therefore, in the reviewer’s opinion, the information about the tyrosinase/melanin pathway should be described widely and a graphical scheme of this path needs to be included in this paper.

2.      In the reviewer’s mind, the author in the background should additionally bring up general information about animals’ cancer immunotherapy, for example:

             adoptive cell transfer – summarized by Bujak et al. https://doi.org/10.1186/s13028-018-0414-4 ;

             monoclonal antibodies - summarized by Beirao et al. https://doi.org/10.1016/j.tvjl.2016.11.005 ;  

             oncolytic virotherapy - summarized by Gentschev et al. https://doi.org/10.3390/v6052122, 

as an introduction to the Oncept vaccine.

3.      Virtually all published studies on this vaccine have touched upon the question of survival and remission time, etc. In the conclusion, the author should include information about the need not only for assessment of tyrosinase expression in tumor before vaccinations but for more basic tests after administration of the vaccine, e.g. antibody titer, the immunophenotype of cells in peripheral blood as well as cytokines concentrations in serum.

Reviewer 3 Report

This manuscript provides a well written summary of the development of the Oncept melanoma vaccine and clinical reports describing the use of the vaccine in dogs and other species.  The summary is thorough and relevant to the field of veterinary oncology and immunotherapy.  There are a few minor concerns listed below that should be addressed before publication of this excellent review.

Lines 42-45:  Sentence "While techniques to target the adaptive immune system have included injection of autologous dendritic cells expanded ex-vivo via transduction with human gp100, a melanoma antigen, as an adjuvant to radiation therapy and an allogeneic whole-cell tumor vaccines expressing various melanoma antigens" is incomplete.  

Lines 53-54:  Sentence "This study evaluated 18 patients and determined that the vaccine was safe and effective based upon T cell responses and prolonged survivals" is not entirely accurate as efficacy could not be concluded definitively based on the design of the study reported in the cited reference.  A quote from the cited reference states "However, patient selection for clinical studies such as these can lead to marked skewing, and it is not possible to draw any conclusions."  Therefore, the statement in this manuscript should be modified to indicate that although vaccine safety and immuogenicity were confirmed, results regarding efficacy were inconclusive.

Line 125:  "Outline" should be edited to "Outlined."

Discussion of the impact of vaccine on outcomes based on historical controls was presented for most but not all studies presented.  For example, discussion of studies by Treggiiari et al. and by Turek et al. made no mention of results regarding the impact of vaccine based on historical controls.  For consistency, a statement regarding the effect of vaccination should be made for each study.  If case numbers were too low or other study limitations were present, a statement should be made indicating that vaccine impact could not be determined. 

Table 1 does not include the study by C.A. Manley et al. 2011.
